# Composition of the Gut Microbiome and Its Response to Rice Stripe Virus Infection in *Laodelphax striatellus* (Hemiptera: Delphacidae)

**DOI:** 10.3390/insects16111135

**Published:** 2025-11-06

**Authors:** Zhipeng Huang, Lu Zhang, Yu Tian, Jiayi Gao, Fang Liu, Yao Li

**Affiliations:** 1College of Plant Protection, Yangzhou University, Yangzhou 225009, China; dx120210138@stu.yzu.edu.cn (Z.H.); zhanglu93@yzu.edu.cn (L.Z.); ty2870305079@163.com (Y.T.); gjy13771140978@163.com (J.G.); 2Jiangsu Co-Innovation Center for Modern Production Technology of Grain Crops, Yangzhou University, Yangzhou 225009, China

**Keywords:** *Laodelphax striatellus*, gut microbiota, rice stripe virus, functional prediction, 16S rRNA sequencing

## Abstract

This study investigates the composition and abundance of gut microorganisms in the small brown planthopper and how they respond to infection with rice stripe virus. The results show that the gut microorganisms of the small brown planthopper are mainly composed of bacteria from three major phyla, which is similar to other insects, but the detailed composition at the genus level is different. When the small brown planthopper carries rice stripe virus, the composition and diversity of the gut microorganisms change: some bacteria decrease, like *Stenotrophomonas*, *Brevundimonas*, and *Brevibacillus*, while others increase, like *Staphylococcus*. Infection with rice stripe virus also alters the potential functions of the gut microorganisms. These results suggest that rice stripe virus reshapes the gut microbiota of the small brown planthopper, thereby deepening our understanding of how viruses, insects, and microorganisms influence each other. This knowledge may open new avenues for controlling insect-borne plant diseases and protecting rice production.

## 1. Introduction

The insect gut harbors a vast reservoir of microorganisms, which exert regulatory functions in various physiological and biochemical processes. For example, the bacterium *Raoultella electrica* performed nitrogen fixation in the leafhopper gut, thus providing nutrients to the host [1]. Gut microbiota in the beetle *Odontotaenius disjunctus*, a wood saprophyte, promoted lignin degradation for absorption by the beetle host [2]. The gut microbiota of the red turpentine beetle, *Dendroctonus valens*, controlled its development by activating the expression of transcription factor 1 (*Hif-1α*) and then regulating D-glucose transport [3]. In *Aedes aegypti*, the high-affinity cytochrome *bd* oxidase produced by the gut bacterium *Escherichia coli* promoted the growth and development of mosquito larvae by inducing molting [4]. In *Drosophila melanogaster*, gut microbes regulate aggressive behavior in host flies through octopamine signaling [5]. In *Apis mellifera*, gut symbionts in the bacterial genus *Lactobacillus* improved the learning and memory ability of honeybees by targeting and modulating tryptophan metabolism [6]. In the parasitoid *Nasonia vitripennis*, the gut bacteria *Serratia marcescens* and *Pseudomonas spp.* degraded the herbicide atrazine and improved wasp survival [7]. *Raoultella terrigena*, a bacterium that resides in the gut of the moth *Thitarodes xiaojinensis*, helped the host degrade the flavonoid quercetin and protected it from polyphenol toxicity [8]. The gut bacterium *S. marcescens* in the fall armyworm, *Spodoptera frugiperda*, significantly reduced the virulence of the entomopathogenic fungus *Beauveria bassiana* [9]. Gut microbes from the cockroach *Blattella germanica* improved resistance to the bacterial pathogen *Salmonella enterica* by regulating the host immune system [10]. In general, insect gut microbes have essential functions in nutrient acquisition, growth and development, behavioral regulation, and resistance to toxic substances and pathogenic microorganisms of insect host.

A large number of studies mainly focusing on blood-feeding insects have shown that the insect gut microbiome has a critical role in viral infection and transmission. Insect gut bacteria associated with viral infection and replication are mainly classified within the phyla Actinobacteria, Proteobacteria, Bacteroidetes and Firmicutes [11]. In *A. aegypti*, the gut symbiont *S. marcescens* secretes SmEnhancin, a compound that degrades mucins bound to epithelial membranes in the midgut, thereby facilitating dengue virus (DENV) infection [12]. Additionally, the gut bacterium *Serratia odorifera* secretes peptides that enhance epithelial permeability, which also promotes DENV infection [13]. Oral administration of *Proteus spp.*, a bacterial gut symbiont, enhanced the expression of genes encoding antimicrobial peptides, thereby inhibiting DENV infection in the epithelial cells of *A. aegyptii* intestines [14]. Furthermore, a *Chromobacterium* strain isolated from *A. aegypti* secreted two types of bacterial lipase that exhibited broad-spectrum antiviral activity against mosquito-borne viruses such as Japanese encephalitis virus (JEV), Zika virus (ZIKV), Sindbis virus (SINV), and yellow fever virus (YFV) [15]. In a strain of wild *Aedes albopictus*, 55 culturable bacterial species were isolated from the gut. Among these, *Rosenbergiella* strain YN46 secreted glucose dehydrogenase, which converted glucose into gluconic acid and induced rapid acidification of the gut environment. This acidification led to inactivation of viral particles and enhanced the resistance of *A. albopictus* and *A. aegypti* to DENV and ZIKV infection [16]. Unfortunately, the effect of gut microbiota on virus infection and transmission in herbivorous insect vectors remains unclear.

*Laodelphax striatellus* (Fallén) (Hemiptera: Delphacidae), which is known as the small brown planthopper (SBPH), is one of the most dominant herbivores in Asian rice ecosystems due to its capacity for long-distance migration, strong cold tolerance, and high outbreak potential [17]. SBPH not only causes yield losses by feeding on rice phloem via its piercing-sucking mouthparts, but also acts as a vector for transmitting destructive plant viruses such as rice stripe virus (RSV) [18]. RSV transmission occurs via both persistent and circulative-propagative means within SBPH populations [18,19]. After feeding on RSV-infected rice, SBPH ingests the virus, which initially colonizes the midgut epithelium; RSV then spreads to the visceral muscle tissue in the SBPH midgut and begins to replicate [19]. The SBPH gut harbors abundant symbiotic bacteria, primarily in the phyla Proteobacteria, Firmicutes, Bacteroidetes, and Actinobacteria [20]. However, whether these gut microorganisms are involved in the infection and transmission of RSV is unclear.

In this study, the composition of the SBPH gut microbiome was investigated by high-throughput amplicon sequencing combined with artificial isolation and culture. The composition and abundance of gut microbiota in viruliferous and naïve SBPH populations were compared to elucidate RSV-induced alterations in the gut microbiome. These findings provide a basis for further studies on the role of the gut microbiota in RSV infection and transmission.

## 2. Materials and Methods

### 2.1. Insect Rearing and Sample Preparation

We collected SBPH originally from the paddy fields in Jiangsu Province and screened the highly viruliferous SBPH. Viruliferous and naïve SBPH populations were obtained and reared on rice seedlings separately. Under same environmental conditions, both noninfected and viruliferous SBPHs were reared independently on 2–3 cm seedlings of rice cv. Wuyujing 3 in glass beakers containing sterile culture medium in a growth incubator at 25 ± 1 °C, with 80 ± 5% RH and a 12 h light-dark photoperiod. At 10-day (d) intervals, planthoppers were relocated to new seedlings to ensure adequate nutrition. A single female mate with a male and feed independently, parents and offspring was monitored as described with RSV-specific antibody (RSV CP antibody) using dot enzyme-linked immunosorbent assay (dot-ELISA) to ensure that SBPH populations were viruliferous [19].

Newly emerged SBPH adults were sampled for sequencing. 50 adults were selected randomly from samples to detect RSV titers in vivo using RT-qPCR based on target gene RSV CP, in order to determine the virus load of the infected adult insects. Viruliferous and naïve SBPH were starved for 12 h to remove residual allochthonous species in the gut prior to extraction. Each SBPH population (*n* = 240 individuals in total, divided into six replicates) was surface- sterilized in 75% ethanol, rinsed in double-distilled H_2_O and air-dried at ambient temperature as described [21]. Samples of SBPH guts were dissected with sterile forceps, homogenized in one milliliter of sterile deionized H_2_O and stored at −80 °C until needed.

### 2.2. Extraction of DNA and PCR Assays

The HiPure Soil DNA Kit was used to isolate total DNA as recommended by the manufacturer (Magen, Guangzhou, China). The hypervariable V3–V4 region of 16S rDNA gene was amplified with primers 341F (CCTACGGGNGGCWGCAG) and 806R (GGACTACHVGGGTATCTAAT). PCR was conducted as follows: 95 °C for 2 min, followed by 27 cycles at 98 °C for 10 s, 62 °C for 30 s, and 68 °C for 30 s; the final step was extension at 68 °C for 10 min. PCR was performed with triplicate samples containing a 50 μL mixture of the following components: 2.5 mM dNTPs, 5 μL; KOD polymerase, 1 μL; primers, 1.5 μL (10 μM); 10× KOD buffer, 5 μL; 25 mM MgSO_4_, 3 μL; and template DNA, 100 ng, with double-distilled water added to a final volume of 50 μL.

### 2.3. Sequencing with Hiseq 2500

Amplified DNA was obtained from agarose gels (2%) with the AxyPrep DNA Gel Extraction Kit as recommended (Axygen Biosciences, Union City, CA, USA). The concentration of each purified amplicon was quantified using a StepOne Plus Real-Time PCR System (Applied Biosystems, Waltham, CA, USA) with a fluorescence-based DNA quantification method to ensure equal library concentrations prior to sequencing. Equimolar ratios of purified DNAs were combined and subjected to paired-end sequencing (2 × 250) on an Illumina platform using established protocols. Raw reads of sequences were then submitted to the NCBI Sequence Read Archive repository under the accession number PRJNA1322143. Each reaction contained three technical replicates.

### 2.4. Sequence Data Processing and Bioinformatics Analysis

FASTP (https://github.com/OpenGene/fastp, accessed on 16 March 2021) was utilized to obtain filtered raw reads, and FLASH (https://github.com/ebiggers/flash, accessed on 16 March 2021) was employed to combine paired-end clean reads as raw tags. The noisy sequences residing in raw tags were filtered with QIIME v. 1.9.1 (https://qiime.org/index-qiime1.html, accessed on 16 March 2021) using quality-filtering conditions described previously [22]. This approach generated clean tags that were subjected to reference-based chimera checking using UCHIME v. r20110519 (http://drive5.com/uchime/uchime_download.html, accessed on 16 March 2021).

After removing chimeras, tags were organized into operational taxonomic units (OTUs) with  ≥ 97% similarity. UPARSE v. 9.2.64 (https://drive5.com/uparse, accessed on 16 March 2021) was used to assess the diversity and abundance of species, and relationships were determined using VennDiagram v. 1.6.16 (https://www.rdocumentation.org/packages/VennDiagram/versions/1.6.16, accessed on 16 March 2021). Typical OTU sequences were categorized as organisms with a Bayesian model, RDP classifier v. 2.2 (https://github.com/rdpstaff/classifier, accessed on 16 March 2021) and SILVA v. 132 (https://www.arb-silva.de/, accessed on 16 March 2021) with a confidence setting of 80%.

The abundance of taxonomic groups was visualized using Krona v. 2.8.1 (https://github.com/marbl/krona/releases, accessed on 20 December 2024), and heatmaps and stacked bar plots were generated with Graphpad Prism v. 8.0.1 (https://www.graphpad.com/updates/prism-801-release-notes, accessed on 20 December 2024). For comparing microbiome composition, alpha diversity indices, including ACE, Chao, Simpson, and Shannon, as well as principal coordinate analysis (PCoA), were calculated using QIIME and we performed permutational multivariate analysis of variance (PERMANOVA). Alpha diversity indices, including ACE, Chao, Simpson, and Shannon, as well as principal coordinate analysis (PCoA), were calculated using QIIME. The similarities and differences in microbial communities among groups were visualized using Circos (http://circos.ca/, accessed on 20 December 2024). Functions of OTUs in the gut microbiota were predicted with the Kyoto Encyclopedia of Genes and Genomes (KEGG) database and PICRUSt v. 2.1.4 (https://picrust.github.io/picrust/, accessed on 20 December 2024). Differences between groups were assessed using Welch’s *t*-test and Wilcoxon rank-sum test as appropriate in R project v. 3.4.1 (https://www.r-project.org/, accessed on 20 December 2024), with statistical significance set at *p* < 0.05.

### 2.5. Enumeration of Microbes in the Gut Microbiota

Target bacterial strains were isolated using a combination of selective and differential culturing strategies including varied incubation conditions and media supplemented with specific carbon sources or selective inhibitors. Specifically, *Stenotrophomonas* was isolated on vancomycin–imipenem–amphotericin B (VIA) agar and incubated at 30 °C for 24–48 h [23], *Brevundimonas* on R2A agar following membrane filtration at 28 °C for up to 7 days [24], *Brevibacillus* on tyrosine agar at 37 °C for 72 h [25], and *Staphylococcus* on Mannitol Salt Agar (7.5% NaCl with mannitol) at 35 °C for 24 h [26]. SBPH guts were dissected in sterile conditions, and 40 individuals were pooled as one biological sample. The dissected guts were placed into centrifuge tubes containing sterile water and steel beads, and samples were homogenized with an ultrasonic grinder at 70 Hz for 120 s. A sterile 96-well plate was prepared, and each well contained 180 μL of double-distilled water; 20 μL of the homogenized sample was added to the wells in the first column and mixed thoroughly by pipetting up and down for a total of 35 times. 20 μL of the mixture from the first column was transferred to the second column, and this gradient dilution process was repeated across the plate. A 2.5 μL sample from each dilution was spotted onto the respective medium and incubated under the specific growth conditions described above. The dilutions where colonies could be accurately counted were selected for enumeration of colony-forming units (CFUs). Colony counts were determined using the corresponding dilution factors to calculate the total viable bacterial load per gut. Statistical differences between groups were analyzed using two-tailed Student’s *t*-tests in Graphpad Prism v. 8.0.1, and differences were considered significant at *p* < 0.05.

## 3. Results

### 3.1. Composition and Diversity of Gut Microbiota in Naïve SBPH

A total of six samples were sequenced as a service provided by Gene Denovo Biotechnology Co., Guangzhou, China (www.genedenovo.com, accessed on 16 March 2021). An average of 68,223 reads per sample were obtained after quality control. Based on 97% sequence similarity, an average of 662 OTUs were identified from reads of naïve SBPH samples, and these represented 23 phyla, 56 classes, 112 orders, 169 families, and 287 genera (Table 1).

Phylogenetic analysis of the gut microbiota in naïve SBPH revealed that bacteria in the phylum Proteobacteria were the most highly represented group (Figure 1A). The Bacteroidetes and Firmicutes phyla were also well-represented; however, Actinobacteria, Verrucomicrobia, Spirochaetes, and Planctomycetes were represented by fewer genera. Krona visualization of the data confirmed that the gut microbiota of naïve SBPH was predominantly composed of bacteria in the phyla Proteobacteria, Firmicutes and Bacteroidetes (Figure 1B). Proteobacteria dominated the gut microbiota and comprised 94.79% of the community structure, whereas Firmicutes and Bacteroidetes accounted for 3.04% and 1.39%, respectively. Alphaproteobacteria and Gammaproteobacteria, which are both in the phylum Proteobacteria, were the most abundant classes. The Rickettsiales was the most highly represented order with a relative abundance of 84%. Among the captured genera, *Wolbachia*, *Stenotrophomonas*, and *Brevundimonas* were the top three genera in terms of relative abundance, and *Wolbachia* was the most dominant genus. To gain a deeper understanding of the distribution of gut microbes in naïve SBPH at the genus level, the bacterial genera in the six different samples were displayed in a heatmap (Figure 1C). In all six samples, *Wolbachia* was the most abundant genus (>1%) followed by *Stenotrophomonas*, *Brevundimonas*, *Brevibacillus*, and *Escherichia-Shigella*. These results indicated that the sequencing depth undertaken in this study was successful in displaying the diverse gut microbiota in SBPH.

### 3.2. Composition and Abundance of Gut Microbiota in Viruliferous and Naïve SBPH

#### 3.2.1. Comparison of Gut Microbiota Composition

Venn diagrams were generated to visually represent the exclusive and shared number of OTUs in the gut microbiota of viruliferous and naïve SBPH (Figure 2A). The viruliferous group exhibited both significantly higher total and unique OTU counts as compared to the naïve group, indicating a greater diversity of gut microbes in viruliferous SBPH. Additionally, we identified a set of 355 core OTUs that were common to both groups. Annotation of the core OTU sequences revealed that the common gut microbes in viruliferous and naïve SBPH samples could be classified into 17 phyla, 25 classes, 52 orders, 74 families, and 102 genera. These core OTUs were used to detect differences in microbial abundance between the two groups.

Multiple gut microbial α-diversity indices, including ACE, Chao, Simpson, and Shannon estimates, were compared in viruliferous and naïve SBPH (Figure 2B). The results showed that all four α-diversity indices of the gut microbiota in viruliferous SBPH were significantly higher than those in naïve SBPH (*p* < 0.05). The structure of the gut microbial community in viruliferous and naïve SBPH was compared by β-diversity analysis based on principal coordinate analysis (PCoA). Gut microbiota from each SBPH population clustered together along the first principal coordinate (PC1), with a highly significant difference in PC1 values between viruliferous and naïve SBPH (*p* < 0.01) (Figure 2C, PERMANOVA test, *R*^2^ = 0.88, *p* < 0.01). These results provide additional evidence that RSV infection markedly alters the gut microbial composition of SBPH.

#### 3.2.2. Comparison of Gut Microbiota Abundance

We examined the relative abundance of the top ten most dominant microbial taxa across all samples to further investigate compositional differences in gut microbiota between viruliferous and naïve SBPH (Figure 2D). At the phylum level, the relative abundance of Proteobacteria, the most dominant phylum, in viruliferous SBPH was 97.44 ± 0.88%, which was significantly higher than that in naïve SBPH (94.79 ± 1.96%) (Figure 3A). In contrast, the relative abundance of Firmicutes, the second most dominant phylum, in viruliferous SBPH was 0.99 ± 0.36%, which was significantly lower than that in naïve SBPH (3.04 ± 1.64%) (Figure 3B). At the class level, the relative abundance of Alphaproteobacteria in viruliferous SBPH (95.27 ± 2.62%) was significantly higher than that in naïve SBPH (87.34 ± 5.01%) (Figure 3C), and the relative abundance of Gammaproteobacteria in viruliferous SBPH (2.12 ± 1.84%) was significantly lower compared to that in naïve SBPH (7.35 ± 2.89%) (Figure 3D). Additionally, the relative abundance of Bacilli in viruliferous SBPH (0.26 ± 0.24%) was significantly lower than that in naïve SBPH (1.97 ± 1.59%) (Figure 3E).

At the order level, Acetobacterales exhibited a dominant presence in viruliferous SBPH with a relative abundance of 76.85 ± 15.16%, which was 55-fold higher than that observed in naïve SBPH (1.38 ± 0.04%) (Figure 3F). Bacteria in the order Rickettsiales showed a markedly reduced presence in viruliferous SBPH (18.15 ± 13.03%), which was 4.6 times lower than in naïve SBPH (83.75 ± 6.66%) (Figure 3G). Similarly, bacteria in the Bacillales were substantially reduced in viruliferous SBPH (0.23 ± 0.21%), which was 8.2 times lower than in naïve SBPH (1.89 ± 1.61%) (Figure 3H).

At the family level, the relative abundance of Acetobacteraceae in viruliferous SBPH was 76.86 ± 5.16%, a level significantly higher than in naïve SBPH (1.37 ± 0.04%) and representing a 56.1-fold increase (Figure 3I). The relative abundance of Anaplasmataceae in viruliferous SBPH (18.14 ± 3.04%) was significantly lower than in naïve SBPH (83.72 ± 6.67%), indicating a 4.6-fold decrease (Figure 3J). Additionally, the relative abundance of Paenibacillaceae in viruliferous SBPH at 0.10 ± 0.18% was significantly lower than in naïve SBPH (1.79 ± 1.63%) (Figure 3K).

The above results indicated that viruliferous and naïve SBPH had extensive differences in the relative abundance of gut microbiota, which can be attributed to the infection of RSV. To gain deeper insight into these differences, we utilized a Circos plot to display the top ten bacterial genera ranked by abundance across all samples, with the number of tags exceeding 2000 (Figure 2E). Among the top four most abundant genera, *Wolbachia* exhibited a significant difference in abundance in viruliferous and naïve SBPH, and its abundance in the gut of viruliferous SBPH individuals was approximately 20% of the abundance in naïve SBPH (Figure 3L).

### 3.3. Comparison of Gut Microbial Loads in Viruliferous and Naïve SBPH

Bacteria that could be artificially cultivated and had different levels of relative abundances based on 16S rRNA high-throughput sequencing were further analyzed to compare gut microbial loads in viruliferous and naïve SBPH. Six culturable genera including *Stenotrophomonas*, *Brevundimonas*, *Brevibacillus*, *Lachnospiraceae_NK4A136_group*, *Staphylococcus*, and *Veillonella* exhibited showed significantly different abundances in viruliferous and naïve SBPH (Figure 4A). Based on these findings, bacteria in four representative genera including *Stenotrophomonas*, *Brevundimonas*, *Brevibacillus*, and *Staphylococcus* were isolated and cultured. Quantitative analyses of these four genera demonstrated that *Stenotrophomonas*, *Brevundimonas*, and *Brevibacillus* exhibited significantly lower CFUs in viruliferous SBPH as compared to the naïve group (Figure 4B–D), whereas *Staphylococcus* had significantly higher CFU counts in the viruliferous group (Figure 4E). Collectively, these culture-based results were consistent with trends observed in sequence analysis. These findings not only validate the reliability of the sequencing data but also provide experimentally tractable bacterial strains for future functional investigations.

### 3.4. Predicted Functions of Gut Microbial Communities in Viruliferous and Naïve SBPH

Predictive functions of gut microbiota in viruliferous and naïve SBPH were obtained using PICRUSt v. 2.1.4 and the KEGG Orthology (KO) database. A total of 29 KEGG pathways were identified in viruliferous and naïve SBPH, with 24 pathways exhibiting significant differences between the two groups. In terms of the cellular processes group, microbial functions related to cell motility, cell growth and death, transport and metabolism, and cell community processes were observed to be significantly higher in viruliferous SBPH than in naïve SBPH (Figure 5A). This suggests that the microbial communities in viruliferous SBPH may be more involved in processes related to cellular dynamics and reflect physiological changes induced in response to RSV. Analysis of metabolic functions revealed that viruliferous SBPH had a significantly higher abundance of microbes involved in carbohydrate, lipid, and amino acid metabolism as compared to naïve SBPH (Figure 5B), suggesting that RSV infection affects microbial-driven metabolic pathways. In genetic information processing (Figure 5C), viruliferous SBPH had a significant increase in microbial abundance in replication/repair, translation, and transcription functions as compared to naïve SBPH. This suggests that viral infection influences microbial genetic processes, possibly through an increase in replication rates to support microbial proliferation in the gut. In the organismal systems category, environmental stress responses and immune system functions were significantly increased in viruliferous SBPH (Figure 5D). Conversely, endocrine system and digestive system functions showed no significant differences between the two groups, indicating that viral infection may not strongly impact the basic endocrine and digestive functions of the microbiome.

We then compared gut microbial functions in response to environmental factors and pathogen infection in viruliferous and naïve SBPH. For environmental information processing, viruliferous SBPH exhibited significant increases in membrane transport and signal transduction when compared to naïve SBPH (Figure 5E). This could indicate enhanced adaptation of the microbial community to the environment within the gut microbiome, potentially as a response to viral infection. Lastly, pathways related to infectious diseases (Figure 5F) exhibited significantly higher microbial abundance in the viruliferous SBPH, but other disease categories were not significantly different in viruliferous and naïve SBPH.

## 4. Discussion

In this study, Proteobacteria, Firmicutes, and Bacteroidetes were the most dominant phyla in the gut microbiota of SBPH, which is similar to results reported by Zhang et al. [20]. These phyla were also the predominant microbial taxa in the microbiomes of *Nilaparvata lugens*, brown planthopper (BPH) [21] and mammals [27]. Our data and other studies indicate that insects including SBPH, *A. albopictus* and *Pyrrhocoris apterus*, have similar intestinal microflora at the class, order and family levels of taxa [28,29]. In the current study, we identified *Wolbachia*, *Stenotrophomonas*, and *Brevundimonas* as the dominant genera in SBPH. A prior investigation reported that *Acinetobacter*, *Wolbachia*, *Serratia*, and *Lactobacillus* are the dominant genera in BPH [21], whereas *Wolbachia*, *Acinetobacter*, *Cardinium*, and *Rickettsia* were dominant genera in *Sogatella furcifera*, the white-backed planthopper (WBPH) [30]. Collectively, this study and prior reports suggest that the core gut microbiota remains relatively conserved, whereas the composition of gut microbiota at the genus level is more variable in different insect species.

Using high-throughput amplicon sequencing, we discovered that RSV significantly affected the abundance of gut microbiota and their potential functions in SBPH. At the phylum level, Proteobacteria underwent a significant increase in viruliferous SBPH as compared to naïve SBPH (Figure 3A). Righi et al. reported that COVID-19 infection increased the abundance of opportunistic bacterial pathogens belonging to the phylum Proteobacteria in the microflora of human patients [31]. Additionally, norovirus infection in humans disrupted the structure of the gut microbiome and led to an increase in Proteobacteria [32], and influenza A virus (IAV) promoted the enrichment of the Proteobacteria population in mice [33]. Collectively, our findings and those previously reported indicate that viral infection increased the abundance of Proteobacteria. Meanwhile, bacterial populations in the taxa Alphaproteobacteria (a class of Proteobacteria) and the family Acetobacteraceae (Figure 3C,I) showed a significant increase in viruliferous SBPH, suggesting that the increased abundance of Proteobacteria in RSV-infected SBPH might be due to increased colonization by Acetobacteraceae. Members of Acetobacteraceae regulated homeostasis in *Drosophila melanogaster* by controlling the speed of development, energy metabolism, body size, and intestinal stem cell activity [34]. Our data suggest that microbial functions in metabolism, cell motility, cell growth and death, and membrane transport were significantly enhanced in the gut of viruliferous SBPH. It is plausible that the increase in the Acetobacteraceae population due to RSV infection may be an attempt to maintain cellular homeostasis. Our speculation requires further investigation.

In contrast, the abundance of two members of Alphaproteobacteria, e.g., the order Rickettsiales (Figure 3G) and the genus *Brevundimonas* (Figure 4C), were significantly reduced by RSV infection. *Wolbachia*, a member of the order Rickettsiales, is known to confer viral resistance in *A. aegypti* and *D. melanogaster*, and resistance was particularly effective against RNA viruses such as DENV and Chikungunya virus (CHIKV) [35,36]. Furthermore, *Wolbachia* was shown to activate the Toll, IMD and Jak/STAT defense pathways in the *A. aegypti* host [37]. Our data indicated that immune-related functions of the gut microbiota were significantly enhanced in viruliferous SBPH (Figure 5D). The correlation between the altered abundance of Rickettsiales in the gut of SBPH and changes in the immune response caused by RSV infection warrant further study.

The population of Gammaproteobacteria, another member of the Proteobacteria, was significantly lower in viruliferous as compared to naïve SBPH (Figure 3D), and the genus *Stenotrophomonas* exhibited a similar reduction (Figure 4B). This suggested that the reduced abundance of Gammaproteobacteria may be primarily due to a significant reduction in the *Stenotrophomonas* population. Interestingly, *Stenotrophomonas maltophilia* was reported to induce systemic resistance to cucumber mosaic virus (CMV) and cucumber green mottle mosaic virus (CGMMV) by inducing the expression of genes encoding antioxidant enzymes in tobacco and cucumber, respectively [38,39]. Furthermore, *S. maltophilia* resists oxidative stress by synthesizing superoxide dismutase, catalase, disulfide isomerase, and methionine sulfoxide reductase [40,41]. Prior studies have shown that antioxidant stress-related functions involve KEGG metabolic and environmental information processing pathways [42,43], and this is supported by our observations on the gut microbiota of viruliferous SBPH (Figure 5B,D). The correlation between alterations in the *Stenotrophomonas* population in the gut microbiota of SBPH and changes in metabolism and adaptation to the environment due to RSV infection warrant further investigation.

Changes in the abundance of Firmicutes in response to viral infection can vary depending on the host species. Mizutani et al. reported that COVID-19 infection led to a significant decrease in the abundance of Firmicutes in the human gut microbiota [44], and Corrêa et al. reported that ZIKV infection triggered a significant reduction in colonization of the gut microbiome by the Firmicutes phylum in mice [45]. In contrast, the abundance of Firmicutes has also been reported to increase due to viral infection. In the gut of *Helicoverpa armigera* larvae, infection with *H. armigera* nucleopolyhedrovirus (HearNPV) triggered a significant increase in the abundance of Firmicutes [46]. In the current study, the population of Firmicutes underwent a significant reduction in RSV-infected SBPH. The abundance of Bacilli, a class of Firmicutes belonging to the order Bacillales, was significantly reduced in viruliferous SBPH (Figure 3E), and the genus *Brevibacillus* exhibited a similar, significant reduction (Figure 4D). We speculate that the decrease in Firmicutes might be caused by the reduced *Brevibacillus* population in viruliferous SBPH. *Brevibacillus laterosporus* enhanced the immune responses of house flies by activating a series of defense-related genes [47]. Wang et al. demonstrated that *B. laterosporus* secretes a novel protein elicitor that activated defense responses in *Nicotiana benthamiana*, thus inhibiting Tobacco mosaic virus (TMV) infection [48]. Interestingly, we found that RSV infection increased the abundance of *Staphylococcus*, another genus in the order Bacillales and phylum Firmicutes (Figure 4E), which can respond to viral infection primarily by modulating the immune response [49,50]. Previous studies have shown that microbiome functions are shared by multiple taxa and exhibit functional redundancy; consequently, the loss or decline of specific taxa can be compensated by functionally related microorganisms [51,52]. We speculate that *Staphylococcus* plays a dominant role in immune system-related functions, and its increased abundance may compensate for the reduction in the *Brevibacillus* population. Whether functional redundancy in the SBPH microbiome helps maintain the RSV infection in viruliferous SBPH is unclear and warrants further research.

In summary, these findings suggest that RSV infection significantly alters the structure of the gut microbial community in SBPH, and this is characterized by a significant increase in the relative abundance of Proteobacteria and a concomitant decrease in Firmicutes. The mechanisms by which Proteobacteria and Firmicutes respond to RSV infection may be attributed to their potential roles in maintaining cellular homeostasis and regulating the host immune system, which awaits further experimental validation. Future studies focusing on how RSV-induced shifts in specific microbial taxa can be modulated and harnessed may provide novel avenues for developing microbiota-based strategies to mitigate RSV infection and transmission.

## Figures and Tables

**Figure 1 insects-16-01135-f001:**
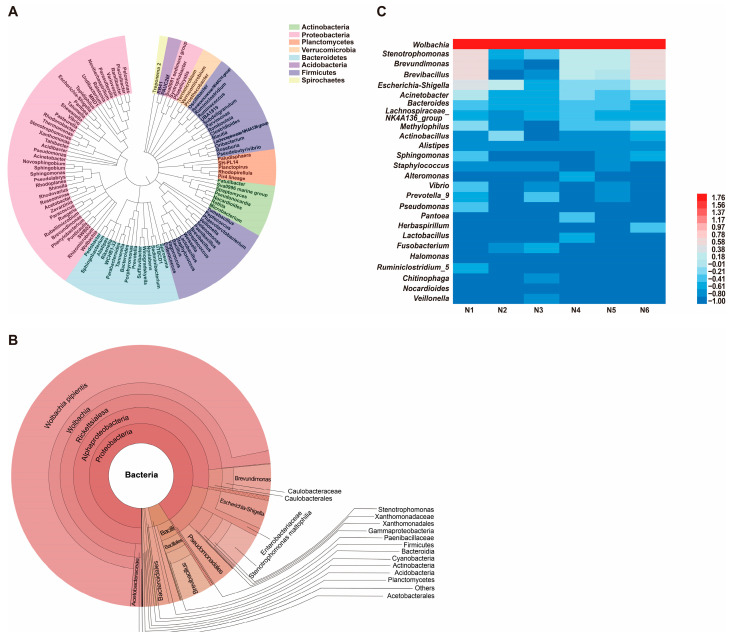
Composition and diversity of the gut microbiota in naïve small brown planthoppers (SBPH). (**A**) Phylogenetic analysis of bacterial genera in the gut microbiota of SBPH. (**B**) Krona analysis of the gut microbiota in naïve SBPH. Results are shown when the proportion of operational taxonomic units was greater than 0.05. (**C**) Heatmap based on the composition and abundance of bacterial genera in the gut microbiota of six naïve SBPH samples, which were designated N1, N2, N3, N4, N5, and N6.

**Figure 2 insects-16-01135-f002:**
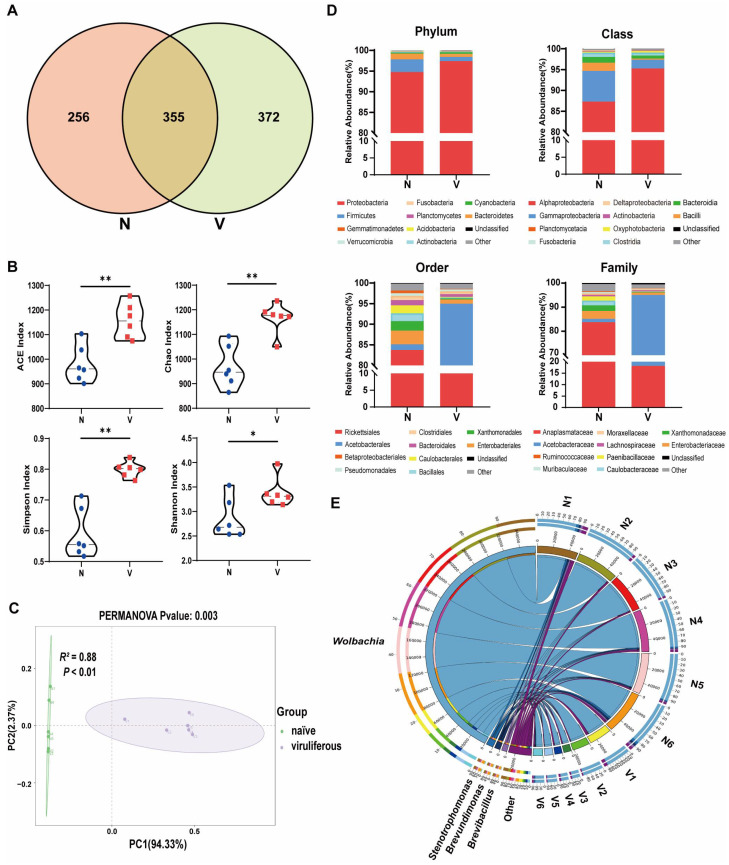
Composition and abundance of gut microbiota in viruliferous and naïve small brown planthoppers (SBPH). (**A**) Venn diagrams showing the comparative distribution of OTUs in naïve and viruliferous SBPH. (**B**) Comparison of ACE, Chao, Simpson and Shannon α-diversity indices in naïve and viruliferous SBPH. (**C**) Principal coordinate analysis of gut microbiota in naïve and viruliferous SBPH. (**D**) Phylum, class, order and family assignments for the top ten most abundant bacterial genera in the gut of naïve and viruliferous SBPH. (**E**) Circos plot of the top 10 most abundant bacterial genera ranked with the number of tags exceeding 2000. Abbreviations: N, naïve; V, viruliferous. Asterisks indicate the following: *, *p* < 0.05; **, *p* < 0.01.

**Figure 3 insects-16-01135-f003:**
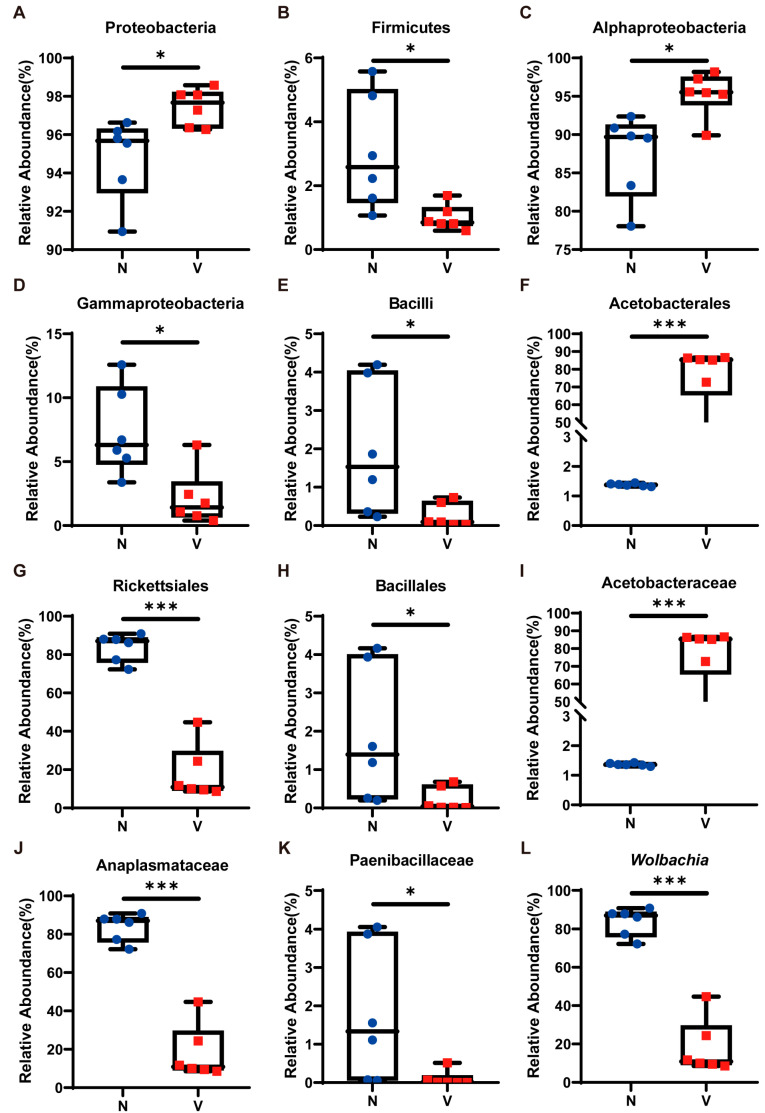
Relative abundance of gut microbiota in viruliferous and naïve small brown planthoppers (SBPH). (**A**) Proteobacteria; (**B**) Firmicutes; (**C**) Alphaproteobacteria; (**D**) Gammaproteobacteria; (**E**) Bacilli; (**F**) Acetobacterales; (**G**) Rickettsiales; (**H**) Bacillales; (**I**) Acetobacteraceae; (**J**) Anaplasmataceae; (**K**) Paenibacillaceae; and (**L**) *Wolbachia*. Abbreviations: N, naïve; V, viruliferous. Asterisks indicate the following: *, *p* < 0.05; ***, *p* < 0.001.

**Figure 4 insects-16-01135-f004:**
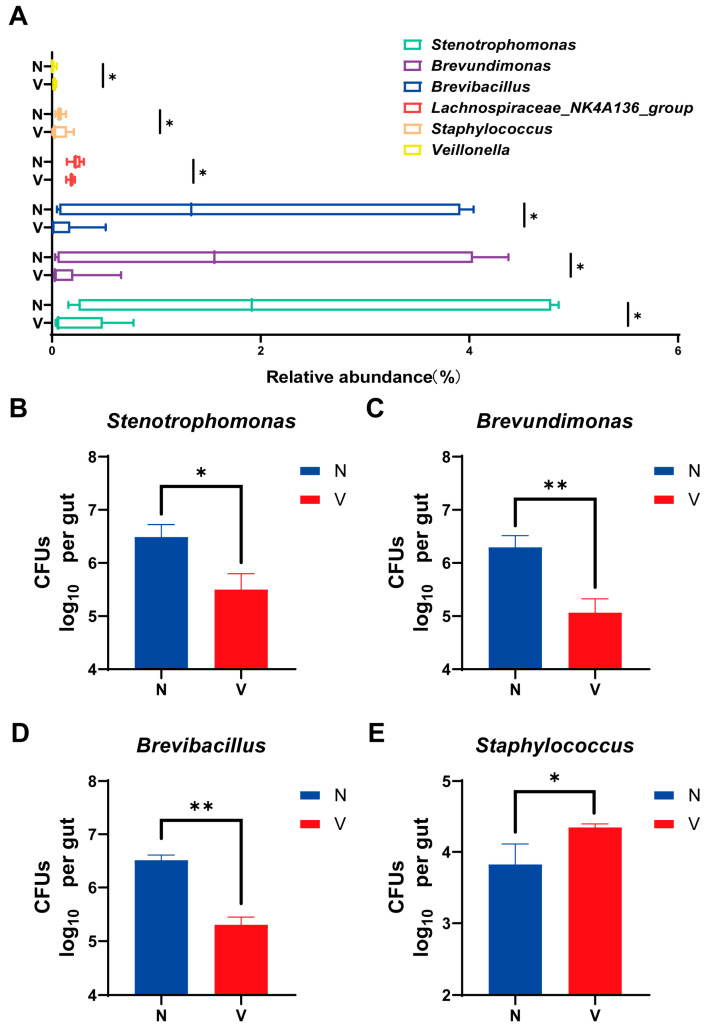
Comparison of gut microbial loads in culturable bacterial genera in the small brown planthopper (SBPH). (**A**) Abundance of different gut microbial genera in viruliferous and naïve SBPH. Panels B-E show colony-forming units (CFUs) of the following bacteria in the gut of SBPH: (**B**) *Stenotrophomonas*; (**C**) *Brevundimonas*; (**D**) *Brevibacillus*; and (**E**) *Staphylococcus*. Abbreviations: N, naïve; V, viruliferous. Asterisks indicate the following: *, *p* < 0.05; **, *p* < 0.01.

**Figure 5 insects-16-01135-f005:**
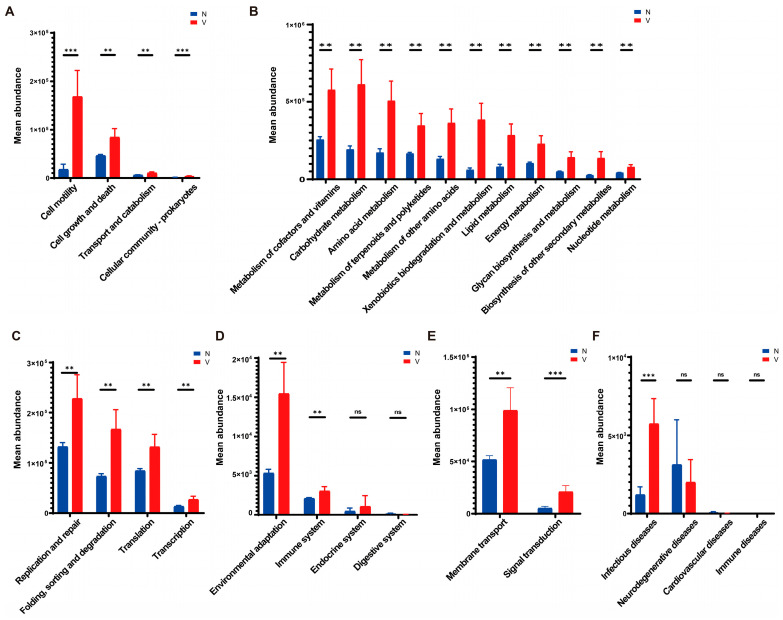
Predictive functions of gut microbiota in viruliferous and naïve SBPH based on PICRUSt software and the KEGG Orthology (KO) database. The functional categories were grouped as follows: (**A**) cellular processes; (**B**) metabolism; (**C**) genetic information processing; (**D**) organismal systems; (**E**) environmental information processing; and (**F**) pathogenicity. Abbreviations: N, naïve; V, viruliferous; ns, not significant. Asterisks indicate the following: **, *p* < 0.01; ***, *p* < 0.001.

**Table 1 insects-16-01135-t001:** Basic information on the 16s rRNA sequencing of the gut microbiota in naïve SBPH.

Samples	Valid Reads	OTUs	Phyla	Classes	Orders	Families	Genera
N1	63,657	736	23	56	112	169	287
N2	69,887	645					
N3	63,972	627					
N4	71,765	664					
N5	68,967	641					
N6	71,092	658					

## Data Availability

The sequencing data generated in this study have been deposited in the NCBI Sequence Read Archive (SRA) under the accession number PRJNA1322143. All other data supporting the findings of this study are included within the article.

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
