# Peer review of "Composition of the Gut Microbiome and Its Response to Rice Stripe Virus Infection in Laodelphax striatellus (Hemiptera: Delphacidae)"

_insects, 2025, doi:10.3390/insects16111135_

Round 1

Reviewer 1 Report

Comments and Suggestions for Authors

This manuscript by Huang et al. investigates the impact of Rice Stripe Virus (RSV) infection on the gut microbiome composition of its primary vector, the small brown planthopper, Laodelphax striatellus. The authors compared the diversity and taxonomic profiles of gut bacteria between healthy and RSV-infected insects. The results of this study provide foundational molecular evidence that the Rice Stripe Virus (RSV) infection directly alters the gut bacteria of its primary insect vector. Through identification of specific microbial taxa, it opens up new avenues for developing novel control strategies to potentially disrupt viral transmission to rice crops.

The data looks promising and significant methodologies have been deployed which give strength to the research. Please address following minor questions:

  1. Before sequencing, how was the successful infection and viral load of the RSV-infected group quantitatively confirmed? Please add detailed method and the target gene used.
  2. Was the insect diet and feeding time strictly controlled and uniform across the healthy and infected groups ?
  3. Line 137-138: It is unclear which housekeeping gene was used for quantification of amplicons. Please explain in details. You may refer to Comparative Ct method by Pfall 2001 in ‘Real-time PCR analysis’ section in publication - Bulletin of Entomological Research. 2017;107(3):281-293. doi:10.1017/S0007485316000961. Please add both the references.

Author Response

Response to Reviewer1

Comments and Suggestions for Authors

This manuscript by Huang et al. investigates the impact of Rice Stripe Virus (RSV)

infection on the gut microbiome composition of its primary vector, the small brown

planthopper, Laodelphax striatellus. The authors compared the diversity and taxonomic

profiles of gut bacteria between healthy and RSV-infected insects. The results of this

study provide foundational molecular evidence that the Rice Stripe Virus (RSV)

infection directly alters the gut bacteria of its primary insect vector. Through identification of specific microbial taxa, it opens up new avenues for developing novel

control strategies to potentially disrupt viral transmission to rice crops.

The data looks promising and significant methodologies have been deployed which

give strength to the research. Please address following minor questions:

Q1: Before sequencing, how was the successful infection and viral load of the RSV

infected group quantitatively confirmed? Please add detailed method and the target

gene used.

A1: RSV not only enters the host plant through the insect saliva secretion, but also is

vertically transmitted to offspring of SBPH. Therefore, we collected SBPH originally

from the paddy fields in Jiangsu Province and screened the Highly viruliferous SBPH.

Subsequently, both viruliferous and naïve SBPH populations were reared on rice

separately. To confirm the infection rate of the viruliferous group, a single female mate

with a male and feed independently, parents and offspring were analyzed via dot

enzyme-linked immunosorbent assay (dot-ELISA) using monoclonal RSV CP-specific

antibodies on a weekly basis. Highly viruliferous colonies were retained and used in

subsequent studies. To determine the viral load of viruliferous colonies for sequencing,

50 adult planthoppers were collected, and RSV titers in viruliferous planthoppers were

detected based on target gene RSV CP using RT-qPCR. We have added detailed

method in line 114-127.

Q2: Was the insect diet and feeding time strictly controlled and uniform across the

healthy and infected groups?

A2: Under same environmental conditions (at 25 ± 1ËšC, with 80 ± 5% RH and a 12-h

light-dark photoperiod in a growth incubator), both noninfected and viruliferous SBPHs

were reared independently on 2–3 cm seedlings of rice cv. Wuyujing 3 in glass beakers

containing sterile culture medium. During the 30-37 day developmental period of

SBPH, both non-infected and viruliferous strains were transferred to fresh seedlings

every 10 days for sufficient nutrition.Q3: Line 137-138: It is unclear which housekeeping gene was used for quantification

of amplicons. Please explain in details. You may refer to Comparative Ct method by

Pfall 2001 in ‘Real-time PCR analysis’ section in publication - Bulletin of

Entomological Research. 2017;107(3):281-293. doi:10.1017/S0007485316000961.

Please add both the references.

A3: We appreciate the reviewer’s thoughtful comment. We would like to clarify that

our study did not involve quantitative real-time PCR (qPCR) for gene expression

analysis. The 16S rRNA gene amplicons were generated solely for sequencing, and the

StepOne Plus Real-Time PCR System mentioned in Section 2.3 was used only to

measure DNA concentration prior to sequencing, not for relative quantification.

Therefore, no housekeeping gene or comparative Ct method (Pfaffl, 2001) was applied.

The instrument was simply employed as a fluorescence-based DNA quantification tool

to ensure equimolar pooling of amplicons for sequencing on the Illumina HiSeq 2500

platform. To avoid any misunderstanding, we have revised Section 2.3 of the Materials

and Methods to clarify this point in line 146-149

Reviewer 2 Report

Comments and Suggestions for Authors

Comments and Suggestions for Authors

The manuscript titled “Composition of the Gut Microbiome and Its Response to Rice Stripe Virus Infection in Laodelphax striatellus” is an interesting approach to understanding how infection with plant viruses affects the composition and diversity of gut microbiota in the SBH. However, some facts are missing from the manuscript.

Main question:

This work aims to answer the following primary questions: can rice stripe virus (RSV) infection modify the makeup of the tiny brown planthopper's (Laodelphax striatellus) gut microbiome, and if so, how may these changes be related to interactions between the virus and its vector? With regard to viruliferous and naïve planthoppers, the authors specifically sought to examine the distinctions in microbial diversity, community structure, taxonomic makeup, and relative abundance. This is done in an effort to determine whether gut symbiotic bacteria are involved in the mechanisms of RSV infection and transmission. It is appropriate for this publication.

Originality:

Introduction: 

Although the introduction offers helpful background information on previously researched virus–insect interactions, it is still too general and fails to highlight the current understanding of the gut microbiota of Laodelphax striatellus (SBPH) or the particular research gap that this study attempts to fill. 16S rDNA sequencing has already been used to thoroughly define the gut microbial community of SBPH in populations gathered from different parts of China. More information about whether alterations in microbial composition are connected to the dynamics and transmission of viral infections should be included in the introduction.

Methodology:

For Methodological Analysis of Microbiome Composition. The PERMANOVA test analysis should be included.

Results and Conclusions:

The manuscript presents a novel discovery, demonstrating that RSV infection is associated with alterations in gut microbial diversity (such as notable variations in alpha diversity indices) and community structure (backed by PERMANOVA and PCoA analysis). Additionally, it pinpoints particular bacterial groups that exhibit either enrichment or reduction following infection, potentially offering insights into the roles that microbes play in virus transmission.

References: References are appropriate

Figures: Figure 5 is informative, but the X-axis font size is too small and should be increased for better readability.

Some correct as follows:

Line 3: add (Hemiptera: Delphacidae)

Line 122,124: typo H2O

Line 254 PERMANOVA test analysis should add in material (Line: 158-168)

Line 321: A total of 29 is better?.

As a result, I believe it can be accepted after some minor comments are improved.

Author Response

Response to Reviewer 2

Comments and Suggestions for Authors

The manuscript titled “Composition of the Gut Microbiome and Its Response to Rice

Stripe Virus Infection in Laodelphax striatellus” is an interesting approach to

understanding how infection with plant viruses affects the composition and diversity of

gut microbiota in the SBH. However, some facts are missing from the manuscript.

Main question:

This work aims to answer the following primary questions: can rice stripe virus (RSV)

infection modify the makeup of the tiny brown planthopper's (Laodelphax striatellus)

gut microbiome, and if so, how may these changes be related to interactions between the virus and its vector? With regard to viruliferous and naïve planthoppers, the authors

specifically sought to examine the distinctions in microbial diversity, community

structure, taxonomic makeup, and relative abundance. This is done in an effort to

determine whether gut symbiotic bacteria are involved in the mechanisms of RSV

infection and transmission. It is appropriate for this publication.

Originality:

Introduction:

Although the introduction offers helpful background information on previously

researched virus–insect interactions, it is still too general and fails to highlight the

current understanding of the gut microbiota of Laodelphax striatellus (SBPH) or the

particular research gap that this study attempts to fill. 16S rDNA sequencing has already

been used to thoroughly define the gut microbial community of SBPH in populations

gathered from different parts of China. More information about whether alterations in

microbial composition are connected to the dynamics and transmission of viral

infections should be included in the introduction.

Methodology:

Q1: For Methodological Analysis of Microbiome Composition. The PERMANOVA

test analysis should be included.

A1: According reviewer’s comment, we added detailed description of PERMANOVA

test analysis in line 172-175.

Results and Conclusions:

The manuscript presents a novel discovery, demonstrating that RSV infection is

associated with alterations in gut microbial diversity (such as notable variations in alpha

diversity indices) and community structure (backed by PERMANOVA and PCoA

analysis). Additionally, it pinpoints particular bacterial groups that exhibit either enrichment or reduction following infection, potentially offering insights into the roles

that microbes play in virus transmission.

References: References are appropriate

Q2: Figures: Figure 5 is informative, but the X-axis font size is too small and should

be increased for better readability.

A2: Thanks for your comment. We have increased the font size of X-axis for better

readability.

Some correct as follows:

Q3: Line 3: add (Hemiptera: Delphacidae)

A3: We added “(Hemiptera: Delphacidae)” in line3.

Q4: Line 122,124: typo H2O

A4: We have corrected the typo H2O

Q5: Line 254 PERMANOVA test analysis should add in material (Line: 158-168)

A5: we added detailed description of PERMANOVA test analysis in line 172-175.

Q6: Line 321: A total of 29 is better?.

A6: “Twenty-nine” was changed to “A total of 29” in line 335.

As a result, I believe it can be accepted after some minor comments are improved.

Reviewer 3 Report

Comments and Suggestions for Authors

The manuscript is comprehensively well-written.

The methods are appropriate for the study, and the figures and results supported the aim of the study.

Although this kind of study has been done in other insect species and organisms, this does not appear like original research but just cataloging what is there to find in naive and viruliferous insect pests.

However, this is a study good to be put out there for the accumulation of studies on the gut microbiome of species.

Some figures are pixilated and can be placed as supplemental material to be viewed separately and clearly.

Author Response

Response to Reviewer 3

Comments and Suggestions for Authors

The manuscript is comprehensively well-written.

The methods are appropriate for the study, and the figures and results supported the aim

of the study.

Q1: Although this kind of study has been done in other insect species and organisms,

this does not appear like original research but just cataloging what is there to find in

naive and viruliferous insect pests.

However, this is a study good to be put out there for the accumulation of studies on the

gut microbiome of species.

A1: We are grateful to the reviewer for their positive evaluation of this study and will

perform in-depth mechanism research in the future.

Q2: Some figures are pixilated and can be placed as supplemental material to be viewed

separately and clearly.

A2: According your comment, we will improve the clarity of our figures to meet the

requirements of the magazine.